# Pressure sensing mat as an objective and sensitive tool for the evaluation of lameness in rabbits

Christin von der Ahe[1,2☉]*, Hannah Marahrens[1,3☉], Michael Schwarze[1], Nina Angrisani[1,2], Janin Reifenrath[1,2]

1 Clinic for Orthopaedic Surgery, Hannover Medical School, Hannover, Lower Saxony, Germany, 2 Lower Saxony Centre for Biomedical Engineering, Implant Research and Development (NIFE), Hannover Medical School, Hannover, Lower Saxony, Germany, 3 Clinic for Small Cloven-hoofed Animals and Forensic Medicine and Ambulatory Clinic, University of Veterinary Medicine Hannover, Hannover, Lower Saxony, Germany

☉ These authors contributed equally to this work.
* vonderAhe.Christin@mh-hannover.de

**Data Availability Statement:** All relevant data are within the paper and its Supporting information files.

## Abstract

In orthopaedic research, the analysis of the gait pattern is an often-used evaluation method. It allows an assessment of changes in motion sequence and pain level during postoperative follow up periods. Visual assessments are highly subjective and dependent on the circumstances. Particular challenge in rabbits is their hopping gait pattern. The aim of the present study was to establish a more objective and sensitive lameness evaluation using a pressure sensing mat. Twelve NZW rabbits were implemented in the study. They got an artificial anterior cruciate ligament transection of the right knee in connection with an experimental study, which investigated PTOA treatment. Rabbits were examined by a visual lameness score. Additionally, load of the hindlimbs was measured by the use of a pressure sensing mat and a video was recorded. Peak pressure and time force integral, defined as cumulated integral of all sensors associated to a hind paw, were evaluated. Preoperative data were collected on three independent days. As postoperative measurement time points, week 1 and week 12 after surgery were chosen. The subjective visual scoring was compared to the objective data of the pressure sensing mat. Following the visual score, lameness in week one was mild to moderate. In week twelve, rabbits were evaluated as lame free bar one. Contrary, following the values of the sensor mat, lameness in week one appeared to be more pronounced and almost all rabbits still showed low-grade lameness in week twelve. Consequently, the pressure sensing mat is more sensitive than the visual score and captures the grade of lameness much more accurately. For specific orthopaedic issues, where subtle differences in lameness are important to detect, the used system is a good supplementary evaluation method.

## Introduction

In orthopaedic research, a wide variety of treatment methods in different animal models have been used in order to make progress in preventing, curing or alleviating the symptoms of

**Funding:** This research was funded by the German Research Foundation (DFG), specific grant number 404534760 (project number). NA received each award. URL to sponsors' websites: https://www.dfg.de/ The funders had no role in study design, data collection and analysis, decision to publish, or preparation of the manuscript.

**Competing interests:** The authors have declared that no competing interests exist.

osteoarthritis (OA) as globally influential disease [1, 2]. Especially the knee joints of sheep, cats and dogs, but also of small laboratory animals such as rodents are often used as a representative model for surgical interventions or injection methods [2–7]. While imaging techniques and histological evaluations focus on morphological criteria of the tissue and structures, the analysis of the gait pattern allows an assessment of changes in the motion sequence during postoperative follow up periods, which can be influenced by individual treatment methods. Additionally, the pain level of the individual animal is reflected in gait patterns [8, 9]. Therefore, different methods have been established to detect changes in gait after interventions on experimental animals, not only in orthopaedics but also in other research domains such as neurological science. Visual assessments using simple descriptions of optical observations, including locomotor rating scales as presented by Basso et al. [10], play a meaningful role, as they are inexpensive, non-invasive and easy to perform by different examiners [8, 11, 12]. However, visual observations and assessments themselves turn out to be highly subjective and dependent on the circumstances [11, 13]. Also, the behavioural pattern of rodents as chosen models hiding pain and lameness due to their nature as prey animals brings more complexity to the assessment by an observer [8, 9]. Thus, observer reliance and sensitivity narrow the validity of visual evaluation results. Video recording using cameras has been described and used to overcome the subjectivity and potentially poor inter-observer consistency of visual lameness scoring systems. They allow to capture contrast differences of paw prints left by rodents running through a tunnel of Plexiglas [14, 15] or reflective markers assigned to bones and joints of the limb to compare jumping trials and joint angles of rabbits pre- and postoperatively [16] with subsequent computer-based assessment. Methods to visualize load distribution of limbs, like manually inking paws of laboratory animals before running over a paper surface [17], got automated using cameras and adequate software programs [4, 14, 15, 18]. Higher reproducibility and collecting of quantified data further enhance the validity of experiments and are benefits of methods that measure ground forces. The detection of ground reaction forces (GRFs) and pressure sensing methods on treadmills were already well established in orthopaedic examination of humans in the last decades [19, 20] and recently transferred to detect changes in gait after interventions on experimental animals like rats, sheep and rabbits [21–24]. Working with rabbits has proven to be convenient due to their non-aggressive and docile natures. Their ease of handling, their short vital cycles and a suitable body size were found beneficial for experimental operations on the distal femur [3, 22, 23]. Rabbits are already used as common animal model for knee injuries [25] providing extended knowledge of knee joint biomechanics, kinetic and kinematic patterns. However, due to the physiologic, hopping gait pattern and their generic behaviour during stress like thumping, objective and reproducible gait analysis of rabbits is challenging and only very few studies are published. Goetz et al. used a high-resolution pressure sensing walkway system to evaluate the progression of post-traumatic osteoarthritis (PTOA) in New Zealand White rabbits and did not find significant differences between operated and non-operated limbs [24]. In a previous preliminary study on PTOA in our group, we found more pathological cartilage changes in non-operated knees of the control group than of the treatment group while differences were not reflected in their gait. However, only a visual scoring was used to assess lameness and a more sensitive method was recommended [26]. Pressure mats are extensively used and their reliability is known in many species [22, 27–29]. Since sensors that fit to the size and weight of rabbits are commercially available, their application in rabbit models seems promising. There has been little research on the simple use of pressure sensing mats to study the weight distribution on the hindlimbs of rabbits until now. The aim was to establish the method as a standardized in vivo method of gait analysis in rabbits for various types of orthopaedic research and others. Therefore, the main hypothesis that lameness assessment with a pressure sensing mat is more sensitive than

visual scoring in rabbit lameness analysis should be verified and aspects of implementation in routine control should be identified.

## Material and methods

### Housing and handling

The in vivo experiments conducted in our institution were authorized according to the German Animal Welfare act and approved by LAVES (Niedersächsisches Landesamt für Verbraucherschutz und Lebensmittelsicherheit (registration number 33.9-42502-04-18/2774). Twelve female New Zealand White rabbits were purchased at the age of seven weeks from Charles River Laboratories, Research Models and Services GmbH, Germany, so they could socialize early on. Growing up to an age of twenty-four weeks with an average bodyweight of 4.08 kg ± 0.45, they were housed in their constant group of 12 animals in constant conditions (RT 18˚C +/- 3˚C, humidity 55% +/- 10%) in ground housing on coarse shavings. The rabbits were able to perform natural behaviours. Their housing area consisted of free space to move freely with different types of hiding places made of wood or cardboard and elevations to jump on. They had constant access to water and pellet food (Kaninchen Haltung, ssniff Spezialdiäten GmbH, Soest, Germany) and were additionally fed with fresh vegetables and autoclaved hay daily. Since they grew up in our house, they got used to human presence and being handled from the age of twelve weeks maximum. Due to them living in a group, these animals never became completely hand-tame, but having started handling them early was meant to achieve a positive impact on the stress level during treatments and the measurement for gait analysis.

One week before surgery, they became accustomed to the full experimental setup for the sensor mat measurements close to the stable room with same climate and lighting conditions by letting the rabbits discover the setup in groups of two to three animals for at least fifteen minutes by themselves several times.

Based on another OA-study rabbits got a surgical intervention at the right knee joint, including mechanical destabilization of the joint by transection of the anterior cruciate ligament (ACLT) and incision in the medial meniscus aiming to achieve the development of post-traumatic osteoarthritis [26].

### Lameness evaluation

To receive a convincing overall picture of the gait changes after surgery, two different methods of gait analysis have been assessed:

1. Visual evaluation of the gait pattern shown in the stable using a clinical scoring

2. Objective measurement of load distribution of the hindlimbs using a pressure sensing mat

**Clinical scoring.** In the postoperative period, animals' weight, general body condition, spontaneous behaviour and wound healing progress were assessed by a score scheme, we just used in a previous study [26]. The pain was evaluated by visual gait analysis, which was performed on each rabbit with a scoring scheme consisting of a point system (Table 1), as previously described [26] in familiar surroundings in the stable room under constant conditions. Clinical scoring was performed daily for the first two weeks and subsequently three times a week. Due to daily work routines, four different experienced veterinarians performed the examinations in turns, all of whom were trained according to the same principles. To maximize inter-observer agreement, scoring at the start of the study was performed together with subsequent spot-check inspections by the principle veterinarian.

**Table 1. Pain score values for the lameness and the orthopaedic examination of the rabbits during the postoperative period.**

| Lameness score | |
| --- | --- |
| 0 | no pathological deviation while standing and in gait pattern |
| 1 | no pathological deviation while standing, low-grade deviation in gait pattern |
| 2 | middle-grade deviation in gait pattern |
| 3 | abnormal position while standing, high-grade deviation in gait pattern (without incident pain) |
| 4 | no load on operated limb while standing or in gait pattern (with pain) |

**Sensor mat measurement.** The square pressure sensing mat we used is commercially available (Novel GmbH, Munich, Germany), model *pliance*® *S2065* with 32 x 32 sensors, 10 x 10 mm each sensor and a measuring frequency of 10 Hz. The sensitive pressure range is 3–240 kPa. The measured signal is transmitted to the *pliance*® *Analyzer PXF 320* (Novel GmbH, Munich, Germany) and to the evaluation computer.

The setup (Fig 1) consisted of a combination of cardboard walls as opaque areas in the back of the room and metal grids as lateral boundaries. This arrangement was meant to achieve a triggered urge to discover and motivation to move around, also ensuring that the examiners could see the animals. Most of the ground floor was covered with a thin artificial green turf to avoid slipping of the paws. Two larger areas of about 2 to 2.5 m² were connected via a passageway with a narrowing as wide as the pressure sensing mat in the form of an hourglass. In the middle of the passage, the pressure sensing mat was placed in a gap of the turf, so the rabbits had no opportunity to hop next to the mat at the sides. This combination was meant to ensure the most natural gait pattern so that changes in the gait may result from the surgical manipulations only.

For each measuring cycle, two randomly chosen rabbits were put into the experimental area. They were given a short time-period (approx. 5 minutes) for orientation and acclimatisation. One examiner entered the setup and guided the rabbits one by one, ensuring that the

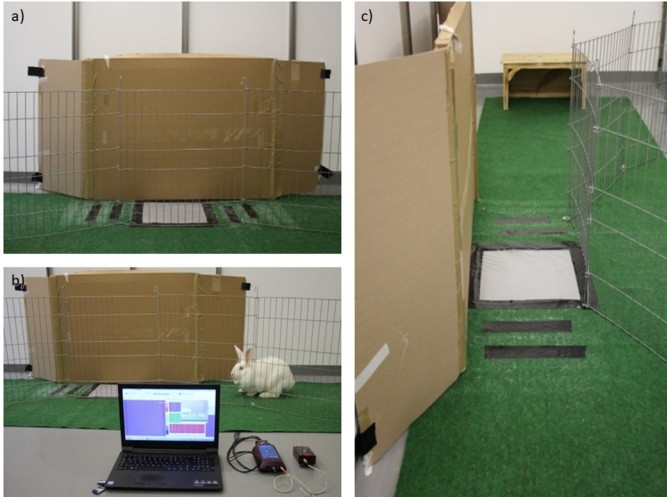

**Fig 1. Experimental setup of gait analysis using the pressure sensing mat from different perspectives.** (a) setup from the side, in the view of an outstanding examiner. (b) perspective of a) complemented by full equipment. (c) frontal perspective with view of the narrowing and the placement of the mat.

animals hit the mat with both hindlimbs. This was achieved by walking slowly behind them, sometimes shooing if they did not move by themselves. Another examiner controlled the measurements via laptop and started recording when announced. The runs were also recorded on video using the integrated webcam, which was placed facing the passing rabbits from the side. Aim was to obtain five successful measurements per animal per day. The evaluation was abandoned ahead of reaching five trials if the rabbit could not or only with considerable effort be motivated to hop over the mat and/or obvious evidence of stress was noticeable. Measurements fulfilling the following criteria were considered successful:

- moving with regular speed without acceleration or braking in the immediate vicinity of the sensor mat

- no hesitation on the mat, hops as evenly as possible

- hops, that did land with both hind paws completely on the mat

## Procedure and scheduling

Prior to surgery, physiological values were determined of each rabbit on three independent days using the sensor mat. After the surgery and a recovery time of seven days, rabbits' gait patterns were examined by sensing load distribution of the hindlimbs once a week for twelve weeks after surgery.

For the comparison of visible lameness score values and data from quantitative measurement with the sensor mat, time points were chosen, where significant lameness occurred (week 1) and lameness was expected to be maximal low grade (week 12).

## Software and data

The software *pliance®-x 32 Recorder* (Novel GmbH, Munich, Germany) was used to link the measurements to the related video records. The region of interest (ROI) of the hind paw prints was manually selected by cropping the time-term and marking specific areas (Fig 2). Data were further processed in a custom R script [30]. To assess the load distribution of the hindlimbs, two parameters were calculated:

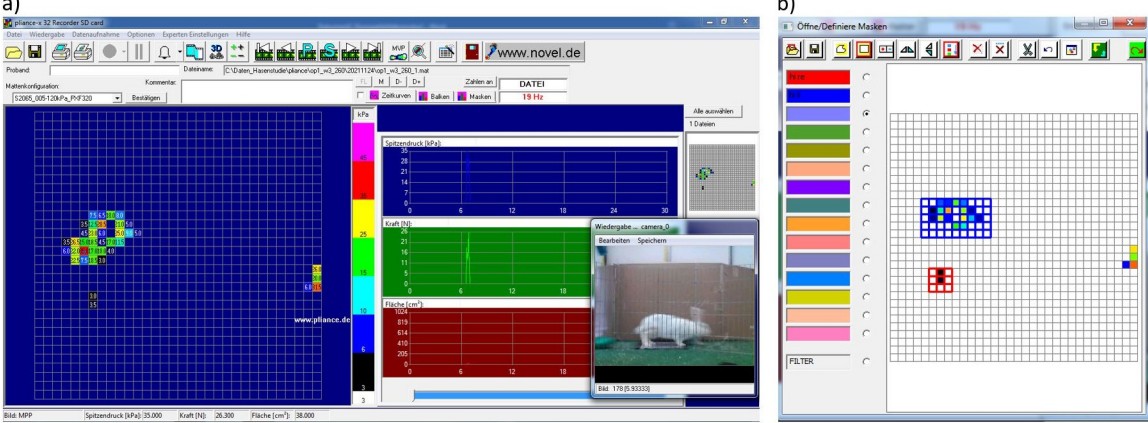

**Fig 2. Exemplary images of *pliance®-x 32 Recorder* software.** (a) entire program interface. (b) example for the selection of ROI (blue: left hindlimb; red: right hindlimb).

1. *peak pressure* (kPa): the highest value of pressure indicated by a single sensor associated to every hind paw contact. Rationale: the rabbits might offload the affected paw in the peak contact pressures,

2. *time-force-integral* (N·s): the integral of force over time during contact by the associated hind paw. Rationale: The rabbits might offload the affected paw by reducing the duration of contact without increasing the force.

To compare the values of peak pressure and time-force-integral, a symmetry Index was calculated ($I = (x_{right}-x_{left})/(x_{right}+x_{left})$) based on Robinson et al. [31]. Values $< 0$ show a higher load bearing of the left limb, while values $> 0$ show a higher load bearing of the right limb. Values $= 1$ imply no load on the left hindlimb and values $= -1$ imply no load on the right hindlimb. In the following, these ratios are named $ratio_{pressure}$ and $ratio_{force}$. Data were exported to Microsoft Excel® (Microsoft Corporation, Redmond, Washington, USA) for further analysis.

## Statistical evaluation

Mean values and standard deviation were determined for metric data. Data for each rabbit at a single measurement day are depicted as boxplot to show variances between single measurements. Mean values of each rabbit per measurement day are compared between different time points with Wilcoxon test because of the small sample (SPSS® version 27, IBM® Deutschland GmbH, Ehningen, Germany). $P < 0.05$ was used as the threshold for significance in all statistical analyses. The range of a physiological run was defined as 95% of all individual values before surgery equally distributed around the median (Q2.5 to Q97.5). No physiological range was defined for the mean values. The natural variance of the runs is insufficiently represented and the low number of values ensure that the range of the physiologically run would be defined too strictly.

## Results

Data are available from all 12 rabbits. For definition of a physiological range for the sensor mat measurement, 160 blank values were collected for the complete group on three independent days (Fig 3). Specific data of the sensor mat measurement are shown in Tables 2 and 3.

In dependence of the measured values, physiological range of the single values was determined between -0.31 and 0.22 ($ratio_{pressure}$) and -0.34 and 0.23 ($ratio_{force}$), respectively.

We met our aim of 5 evaluable runs in 23 out of 60 cases (preoperative 20/36, week 1 0/12, week 12 3/12). However, apart from 3 exceptions where only 2 runs could be evaluated (week 12), we were always able to evaluate at least 3 runs (3 or 4 runs: preoperative 16/36; week 1: 12/12, week 12: 6/12). The most common reason, why individual runs could not be evaluated, was the overlap of individual sensors from different paws in the same time frame and therewith no clear determination of the specific ROIs.

Prior to surgery, the lameness of all rabbits was visually scored with 0. All 12 rabbits showed lameness one week after surgery, which we scored with 1 (n = 6) and 2 (n = 6). In week twelve only one animal was scored with 1, all others were scored as lame free (score 0) (Fig 4a). The lameness preoperative compared to week one and from week one to week twelve show a significant difference (*p < 0.01*). There is no significant difference between preoperative lameness and week twelve (*p > 0.05*). Mean value data of the pressure sensing mat measurement are shown in Fig 4b and 4c. 2 mean values of $ratio_{pressure}$ in the first week (16.7%) and all mean values in the twelfth week (100%) are in the physiological range. 1 mean value of $ratio_{force}$ in the first week (8.3%) and 6 mean values in the twelfth week (50%) are in the physiological range.

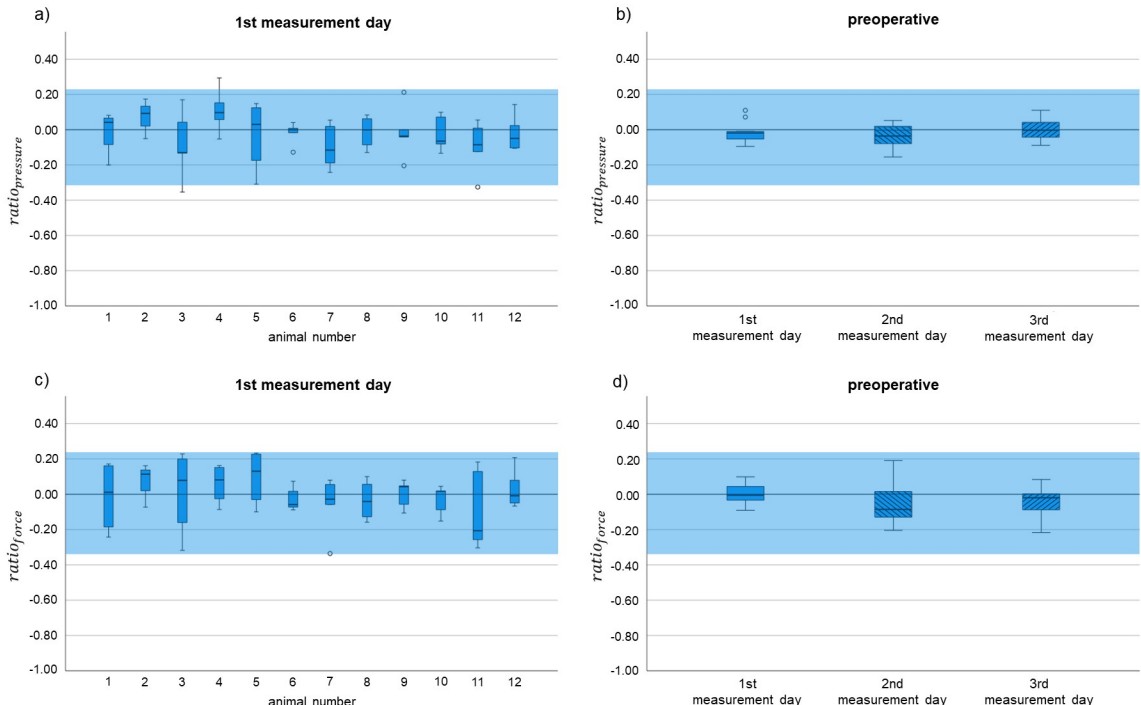

**Fig 3. Preoperative values for *ratio*$_{pressure}$ and *ratio*$_{force}$.** (a) *ratio*$_{pressure}$ of all animals on the first measurement day preoperatively (single values of each rabbit on the exemplary first measurement day). (b) mean values of single rabbits per measurement day depicted as boxplots for *ratio*$_{pressure}$ on the three measurement days preoperatively. (c) *ratio*$_{force}$ of all animals on the first measurement day preoperatively (single values of each rabbit on the exemplary first measurement day). (d) mean values of single rabbits per measurement day depicted as boxplots for *ratio*$_{force}$ on the three measurement days preoperatively.

**Table 2. Pressure sensing mat data of each measurement day of *ratio*$_{pressure}$.**

| measurement day | single values | | | mean values | | |
|---|---|---|---|---|---|---|
| | *range* | *mean value* | *standard deviation* | *range* | *mean value* | *standard deviation* |
| *1ˢᵗ measurement day* | -0.35 to 0.29 | -0.02 | 0.13 | -0.09 to 0.11 | -0.02 | 0.06 |
| *2ⁿᵈ measurement day* | -0.31 to 0.35 | -0.04 | 0.15 | -0.15 to 0.05 | -0.04 | 0.06 |
| *3ʳᵈ measurement day* | -0.31 to 0.24 | 0.00 | 0.14 | -0.09 to 0.11 | 0.00 | 0.06 |
| *preoperative (all 3 measurement days)* | -0.35 to 0.35 | -0.02 | 0.14 | -0.10 to 0.05 | -0.02 | 0.04 |
| *week 1* | -1.00 to 0.15 | -0.56 | 0.28 | -0.95 to -0.10 | -0.54 | 0.25 |
| *week 12* | -0.46 to 0.20 | -0.18 | 0.15 | -0.29 to -0.01 | -0.17 | 0.09 |

**Table 3. Pressure sensing mat data of each measurement day of *ratio*$_{force}$.**

| measurement day | single values | | | mean values | | |
|---|---|---|---|---|---|---|
| | *range* | *mean value* | *standard deviation* | *range* | *mean value* | *standard deviation* |
| *1ˢᵗ measurement day* | -0.34 to 0.23 | 0.00 | 0.14 | -0.09 to 0.10 | 0.00 | 0.05 |
| *2ⁿᵈ measurement day* | -0.57 to 0.30 | -0.05 | 0.19 | -0.20 to 0.19 | -0.05 | 0.11 |
| *3ʳᵈ measurement day* | -0.47 to 0.25 | -0.04 | 0.16 | -0.22 to 0.08 | -0.04 | 0.08 |
| *preoperative (all 3 measurement days)* | -0.57 to 0.30 | -0.03 | 0.16 | -0.10 to 0.05 | -0.03 | 0.05 |
| *week 1* | -1.00 to -0.16 | -0.79 | 0.22 | -1.00 to -0.31 | -0.78 | 0.22 |
| *week 12* | -0.63 to 0.06 | -0.33 | 0.19 | -0.56 to -0.09 | -0.33 | 0.16 |

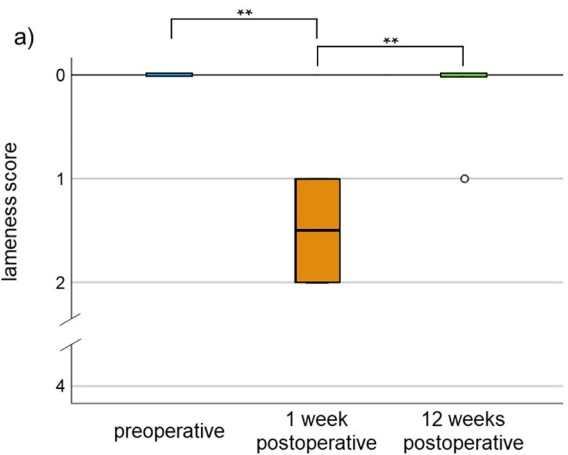

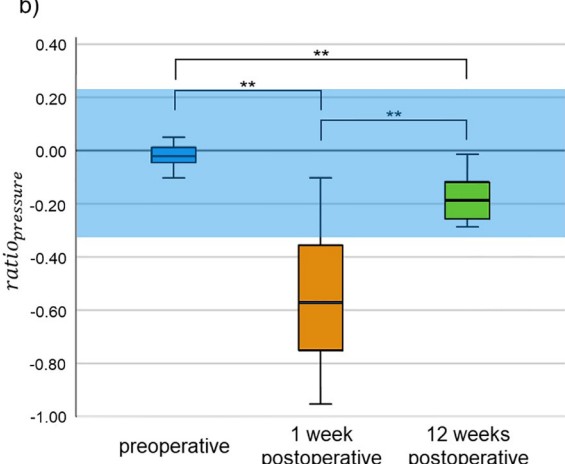

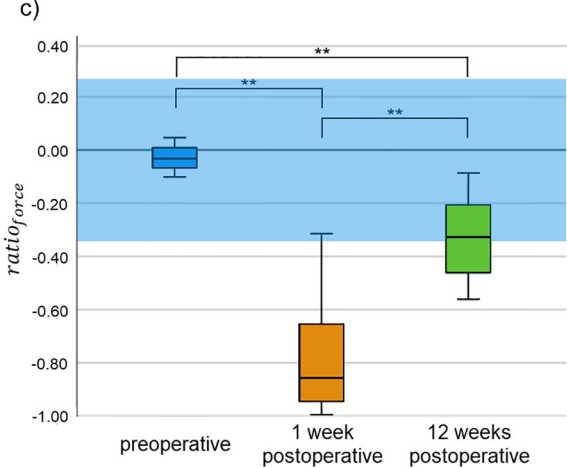

**Fig 4. Lameness score and mean values of $ratio_{pressure}$ and $ratio_{force}$ preoperative, 1 week and 12 weeks postoperative.** Lameness was detected 1 week after surgery in all used evaluation parameters. Whereas rabbits seemed lame free with one exception after 12 weeks in the visual scoring (a), sensor mat measurement assessed a significant lameness in both examined parameters ($ratio_{pressure}$ and $ratio_{force}$) after 12 weeks (b, c). ** = p < 0.01. [a]blue area = defined physiological range.

Compared to the lameness score, mean values of $ratio_{pressure}$ and $ratio_{force}$ in week twelve show a significant difference to the mean values before surgery ($p < 0.01$) (Fig 4b and 4c).

For comparative analysis, individual visual lameness scores in week one and twelve after surgery were illustrated to the measured values of the sensor mat (Fig 5). 7 individual values of $ratio_{pressure}$ in the first week (17.5%) and 25 individual values in the twelfth week (56.8%) are in the physiological range. 4 individual values of $ratio_{force}$ in the first week (10%) and 26 individual values in the twelfth week (59.1%) are in the physiological range.

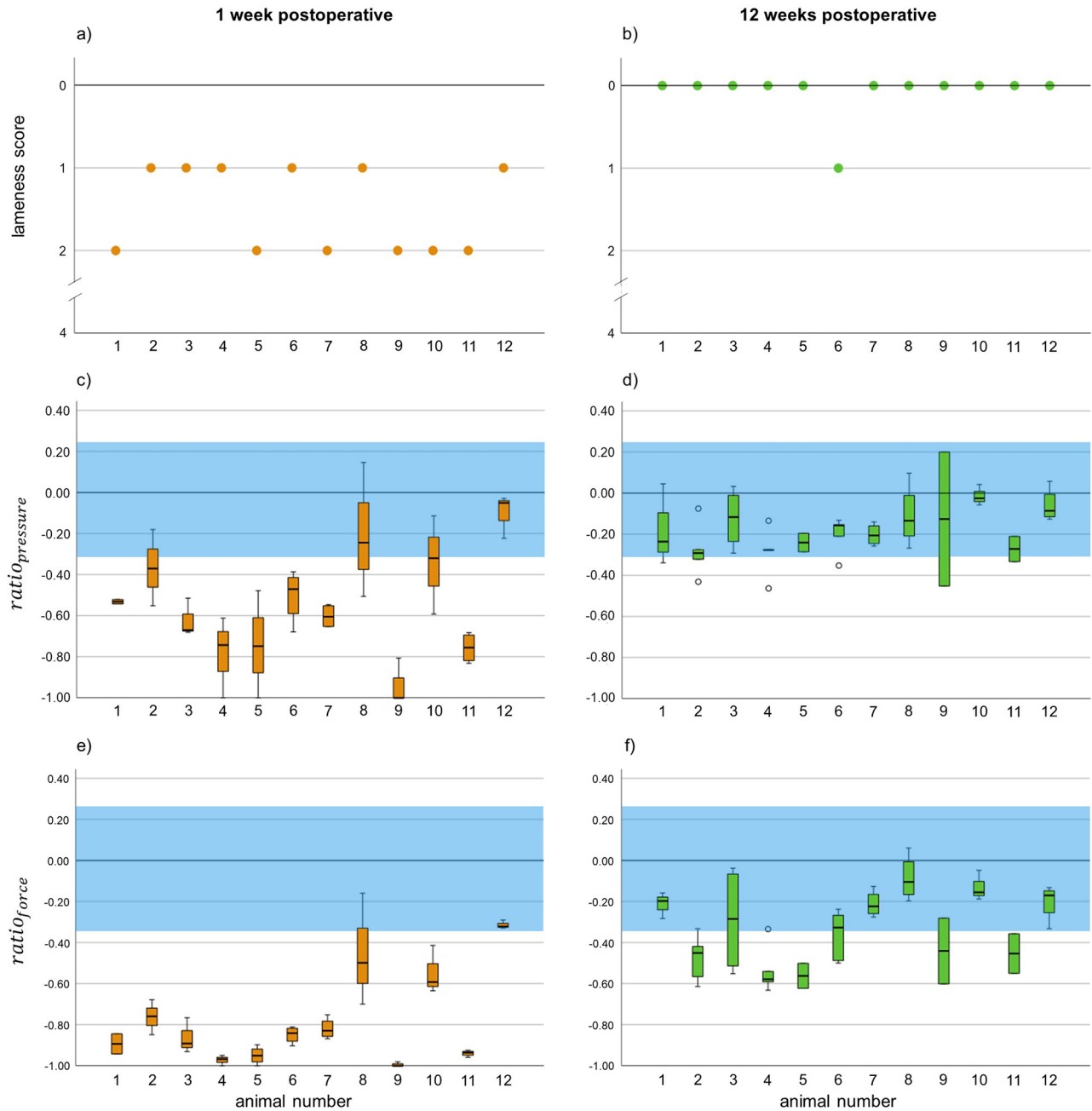

**Fig 5. Individual values of lameness score, $ratio_{pressure}$ and $ratio_{force}$ of all animals 1 week and 12 weeks postoperative.** [a]blue area = defined physiological range.

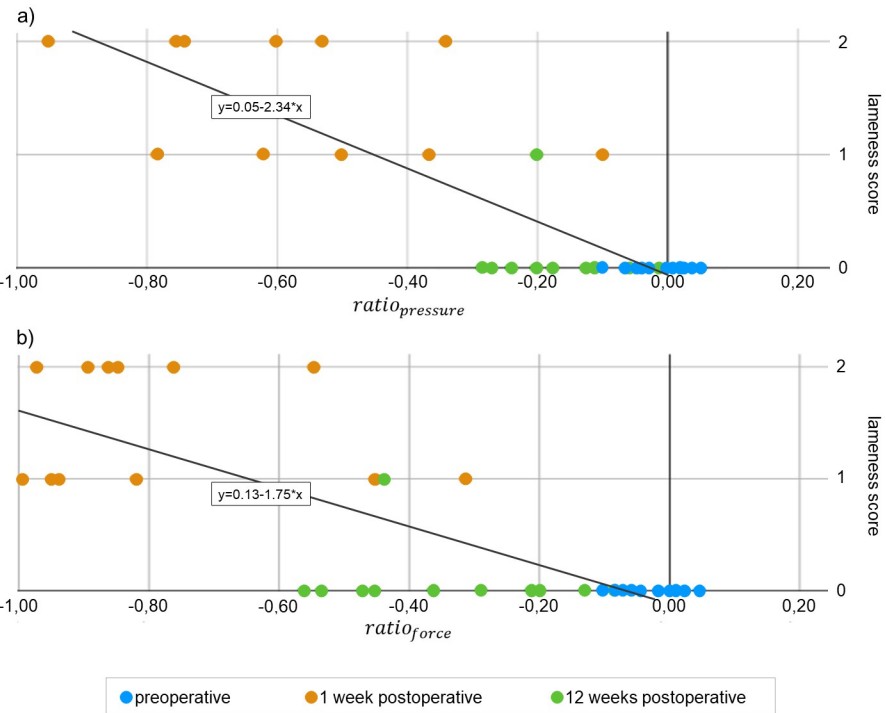

**Fig 6. Scatterplot of lameness score with *ratio_pressure* and *ratio_force*.** (a) $R^2$ = 0.669 (p < 0.001); ρ = -0.743 (p < 0.001). (b) $R^2$ = 0.622 (p < 0.001); ρ = -0.752 (p < 0.001).

A direct correlation of single animals is shown in Fig 6. The correlation coefficient $ρ_{pressure}$ amounts to -0.743 and $ρ_{force}$ = -0.752, respectively. The coefficient of determination $R^2_{pressure}$ amounts to 0.669 and $R^2_{force}$ = 0.622, both with a significance < 0.001. While rabbits in week one with a score of 1 contemporary showed mean values of *ratio_pressure* between -0.78 and -0.10, rabbits with a score of 2 showed mean values of *ratio_pressure* between -0.95 and -0.34. In week twelve, one rabbit has a score of 1 and contemporary a mean value of *ratio_pressure* of -0.20. This mean value is almost classified in the middle range of the other mean values from week twelve (Fig 6a). The values of *ratio_force* are similar in trend (Fig 6b). Data present that visual scoring underestimates the objective lameness evaluation.

## Discussion

In this study, we evaluated the practicability of a pressure sensing mat to assess load sharing of hindlimbs as evidence for lameness in rabbits. We compared the results of the measurement to routinely used lameness score with focus on objectivity and sensitivity.

The hypothesis that lameness assessment with pressure sensing mat is more sensitive than visual scoring in rabbit lameness analysis could be proven. Pressure sensing mat shows differences to preoperative physiological gait that the score does not show. Additionally, degree of lameness seems to be assessed differently in the visual scoring compared to the sensor mat measurement. In week 1 and 12, the sensor mat detected a wide variance in the degree of lameness whereas in the visual score no animal exceeded the score value of 2, which is classified as a middle deviation in gait pattern. Taking into account that a ratio of -1 is equivalent to no load bearing of the affected limb at all, at least the animal with *ratio_pressure* of -0.95 would have been expected to be minimally assessed with a score of 3 in visual examination. Generally, the

assignment to score 1 or 2 cannot be predicted. In week 12, the trend is similar, but not as wide. $ratio_{force}$ shows the same trend. One animal with a visual score of 0 reached a value of -0.56 on the sensor mat, which represents a considerably reduced load on the operated limb. The visual scoring system in the present form led only to three expressions, even in animals that barely loaded the affected limb.

This lower sensitivity of the lameness score is also evident when comparing the respective values between the investigated weeks. While the results of the sensor mat could determine significant differences for both parameters in all weeks, the score values show a significant lameness only in week 1 but almost no lameness in week 12. No significant differences between preoperative scoring and scoring in week 12 can be found. Nevertheless, the correlation coefficient and the coefficient of determination generally show that the lameness score follows the same trend as the sensor mat data. For lameness analysis in experimental studies visual scoring is routinely used in different animal models, e.g. rodents, horses, pigs and dogs [32–37]. In these studies, the gait analyses rather served to compare different analgesics or different osteo-arthritis-inducing methods than to assess the load on the limbs in general. Mostly, the visual assessment was combined with other methods, like kinematic or kinetic gait analysis. Two of these studies just used rough visual assessment of gait and stance, and this appeared to be sufficient for detecting analgesic effect of substances [35, 37]. In terms of animal welfare, in our setup, the lameness can be assessed sufficiently for adapting pain medication. However, visual observations and assessments themselves turn out to be highly subjective and dependent on the circumstances [11]. A limitation of this study is that different veterinarians scored the animals on the different days, thus potentially leading to inconsistent score values. Validity would be considerably enhanced if all experimenters would have scored at each day followed by subsequent calculation of the inter-observer agreement. Due to technical reasons, this was not possible. However, inter-observer agreement was maximized by I. including only veterinarians with experience in rabbit lameness evaluation, II. thorough training under supervision of the principal veterinarian at the start of the study and III. spot-check inspection of consensus by the principal veterinarian during the course of the study. In contrast, objective differences can be detected with the pressure sensing mat, especially there is no direct influence of the examiner. Therefore, the sensor mat indicates the hindlimb loading ratio as a sign of lameness reliably and sensitively.

Our analysis shows variances of both the individual animal and individual measurement days. Preoperatively, only a small scatter is seen in the mean values on the 3 independent measurement days. The standard deviations are all similar. Mainly, week 1 shows a rather wide dispersion of the values. Basically, it is noticeable that the standard deviations of $ratio_{force}$ are higher than those of $ratio_{pressure}$. While $ratio_{pressure}$ only includes the data of a single sensor, $ratio_{force}$ includes the total of all contacted sensors. Thus, it would be logical if the scatter of the data was higher for $ratio_{pressure}$. The influence of time in $ratio_{force}$ could cause a larger variance of data. However, this should be largely eliminated by the ratio. We have not found a precise explanation for this standard deviation difference. Comparatively, in the single values between the individual animals vary considerably with e.g. maximum differences between animal 1 and 8. Independent of the different scores or sensor mat values obtained in week 1 and 12, there was no obvious difference in motivation to run for the single animals. The total number of all runs on these measurement days was similar. Nevertheless, the standard deviation of the objective sensor mat data in week 1 and 12 indicate nonuniform loadbearing. To our knowledge, such an evaluation of rodent limb loading has not been performed to date. Furthermore, gait analysis was mostly performed using relative weight loading of the limb relating to the current body weight [38]. However, in the study by Voss et al., in which this form of GRF

measurement was used, it is noticeable that standard deviations are also larger in animals which are impaired in gait due to injury or pain [38].

About the quality of individual runs, little is reported. In the case of overlapping sensors of the front and hind paws, we did not evaluate the run. Kim et al., who performed gait analysis in sheep, copied such a measurement and performed the calculation once only for the forelimbs and once only for the hindlimbs, so that the run could still be evaluated [39]. However, like us, they did not include measurements in the evaluation where the animal ran unsteadily. We also tried to create a best natural hop by allowing the rabbits to run freely across the mat. However, as described in the methods, rabbits might need to be motivated by the experimenter to hop which could potentially influence the gait pattern. Beside the already referenced literature, there are numerous papers, which report on different gait analysis systems using either voluntary movement as in open fields or catwalk systems, or forced movement as in automated systems using treadmills [40–43]. Petkova et al. are one of the few references who used different gait analysis techniques for the same study [40]. However, they only report on the different parameters which resulted from the regarding technique but did not comment on advantages and disadvantages of any method.

The review by Deuis et al. succeeds in presenting different setups for the evaluation of pain including a paragraph on weight bearing and gait analysis [41]. They also assess the different advantages and disadvantages. Although they point out the value of unrestrained pain evaluation in animals, they do not comment on the impact of forced motion. Although we also believe, that unrestrained gait analysis would be superior to the here presented setup, realization in a rabbit model as used here is difficult. While treadmills are used for rabbits in studies including cardiovascular subjects, automated treadmills with cameras for gait analysis as used for mice and rats are not established yet for the rabbits. The group by Juncosa et al. used a treadmill, which is commonly used in humans with further adaption to make it "animal safe" using a custom-designed, clear, polycarbonate enclosure [23]. The rabbits only performed three hops inside this enclosure, of which the middle one was chosen to evaluate. Although having a constant treadmill speed, the rabbits themselves did not move at a constant speed. Furthermore, no ground reaction forces could be measured since the animals did not like to hop on the Tekscan measuring mat [23]. In our study, after prompting, the veterinarian did not affect the run itself. If the gait pattern over the mat was not continuously (e.g. stopping, or stamping with one hind paw or similar events), the run was not included in the calculation but repeated. Therewith, we assess the influence of this moving stimulus as negligible.

Basically, it would have been advantageous if our sensor mat had been longer to avoid jumping over it and to have the possibility of selecting one of several hoppers in a run. Though, this is also associated with significantly higher costs. Thus, the measurement time and the number of runs for the rabbits might have been shorter and more. In other studies, the researcher aimed for 4, 5, and 10 successful runs [33, 38, 39, 44, 45].

To be able to determine how a physiological run of a rabbit is defined, preoperative measurements were performed on 3 days to calculate a physiological range. In our study this range is very slightly shifted into the negative area. That means our animals had already little more load the left hindlimb preoperatively. The rabbits ran across the mat in both directions and the floor was flat. A cornering was also not detected and the rabbits hopped evenly across the mat without hesitating or stopping. Despite this small shift, there is a clear postoperative lameness.

The acclimation to the setup, which we performed before starting the experiment, should also serve to make the animal run evenly and naturally. However, we had no comparison to the running behaviour without acclimation before the measurement period. Other researchers have also done this in advance [23, 25, 39]. Kim et al. noticed that despite good acclimation,

sheep suffer from stress because they are separated from the flock during the measurement runs [39]. We noticed that care should be taken not to place high-ranking and low-ranking animals together in the setup. In contrast to measuring with the mat, it is possible to perform visual scoring in the stable. The animals can remain in the familiar environment and only need to be accustomed to the general handling.

From a practical perspective, filming the runs turned out useful. It is possible to recognize a hesitation in front of the mat or an unsteady run, respectively, which could falsify the measurement. By considering only the mat measurement, this assessment is barely possible. However, cornering is difficult to judge by filming from the side. In a right-hand curve the left hindlimb may be loaded more than during a straight run. Thus, the rabbit would show more lameness as it really is. Therefore, it might be useful to film the run from two directions, for example from above. To film the animal from the side and the pawprints from below additionally, Allen et al. used a video camera lateral to the setup and a mirror [45]. Video cameras positioned laterally and longitudinally in another study, to see the walking animal from different points of view [46]. The use of different recording perspectives might be a further improvement of the applied setup in the current study. However, it should be kept in mind to provide a method which is easy to use and which can be implemented in experimental studies without substantial prerequisites.

## Conclusion

As there has been little research on the use of pressure sensing mats to study the load distribution on the hindlimbs of rabbits till now, the work presented here is focused on sharing observed benefits and disadvantages during lameness evaluation in experimental studies.

Our results clearly show that the pressure sensing mat measures the grade of lameness more accurately than the visual score. In addition, the values are objectively determined and thus do not depend on different individual assessment of observers. The parameters *peak pressure* and *time-force-integral* seem to be both suitable for lameness evaluation. It is useful to measure the animals' gait prior to surgery, as animals might differ individually in their load distribution between the limbs.

Since this method is non-invasive and the measurement system is portable, measurements can be performed anywhere. However, it is important to make sure that habituation can occur.

Lameness analysis using a pressure sensing mat is a good way to detect small differences in lameness, for example, to elicit the difference between analgesics or surgical methods.

## Supporting information

**S1 Table. Minimal data set underlying the results.**
(PDF)

## Acknowledgments

The authors thank Merle Kempfert, Heidi Harting and Diana Strauch for excellent assistance in performing the experiments and Holger Neumann from novel GmbH for great helping with software problems.

## Author Contributions

**Conceptualization:** Nina Angrisani, Janin Reifenrath.

**Formal analysis:** Christin von der Ahe, Michael Schwarze.

**Funding acquisition:** Nina Angrisani.

**Investigation:** Christin von der Ahe, Hannah Marahrens, Michael Schwarze, Nina Angrisani, Janin Reifenrath.

**Methodology:** Michael Schwarze, Nina Angrisani, Janin Reifenrath.

**Project administration:** Nina Angrisani, Janin Reifenrath.

**Validation:** Christin von der Ahe.

**Writing – original draft:** Christin von der Ahe, Hannah Marahrens.

**Writing – review & editing:** Michael Schwarze, Nina Angrisani, Janin Reifenrath.

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
