## [Decision Letter · Decision Letter 0]

1 Feb 2023

PONE-D-22-32303Pressure sensing mat as an objective and sensitive tool for the evaluation of lameness in rabbitsPLOS ONE

Dear Dr. von der Ahe,

Thank you for submitting your manuscript to PLOS ONE. After careful consideration, we feel that it has merit but does not fully meet PLOS ONE’s publication criteria as it currently stands. Therefore, we invite you to submit a revised version of the manuscript that addresses the points raised during the review process. Please submit your revised manuscript by Mar 18 2023 11:59PM. If you will need more time than this to complete your revisions, please reply to this message or contact the journal office at plosone@plos.org. Please include the following items when submitting your revised manuscript:A rebuttal letter that responds to each point raised by the academic editor and reviewer(s). You should upload this letter as a separate file labeled 'Response to Reviewers'.A marked-up copy of your manuscript that highlights changes made to the original version. You should upload this as a separate file labeled 'Revised Manuscript with Track Changes'.An unmarked version of your revised paper without tracked changes. You should upload this as a separate file labeled 'Manuscript'.

We look forward to receiving your revised manuscript.

Kind regards,

James H-C. Wang

Academic Editor

PLOS ONE

Journal Requirements:

2. Thank you for including your ethics statement:  The study was carried out in strict accordance to the German Animal Welfare act (registration number 33.9-42502-04-18/2774).

To comply with PLOS ONE submissions requirements, please provide the following information in the

Methods section of the manuscript and in the “Ethics Statement” field of the submission form (via “Edit

Submission”):

Please confirm whether an animal research ethics committee prospectively approved this research. Please also enter the name of your Institutional Animal Care and Use Committee (IACUC) or other relevant ethics board. Also include an approval number if one was obtained.

4. Please ensure that you refer to Figure 2 in your text as, if accepted, production will need this reference to link the reader to the figure.

5. We note you have included a table to which you do not refer in the text of your manuscript. Please ensure that you refer to Table 3 in your text; if accepted, production will need this reference to link the reader to the Table.

Additional Editor Comments:

Specifically, the reviewers have carefully reviewed your manuscript. They have also provided their thoughtful comments and suggestions, many of which require clarifications with the goal to improve the clarity of the manuscript. This editor concur with reviewers' critiques, and encourages the authors to consider them closely and wherever pertinent, to revise the manuscript accordingly.

Reviewers' comments:

Reviewer's Responses to Questions

**Comments to the Author**

1. Is the manuscript technically sound, and do the data support the conclusions?

Reviewer #1: Yes

Reviewer #2: Partly

2. Has the statistical analysis been performed appropriately and rigorously? 

Reviewer #1: Yes

Reviewer #2: Yes

3. Have the authors made all data underlying the findings in their manuscript fully available?

Reviewer #1: Yes

Reviewer #2: Yes

4. Is the manuscript presented in an intelligible fashion and written in standard English?

Reviewer #1: Yes

Reviewer #2: Yes

5. Review Comments to the Author

Reviewer #1: Pressure sensing mat as an objective and sensitive tool for the evaluation of lameness in rabbits

The considered paper describes the use of pressure-sensitive mats to evaluate lameness in rabbits.

The title is appropriate for when it is explained in the paper.

The keywords are incomplete, I would add "rabbit" to improve the search possibilities

The biomechanics of the rabbit's gait complicates the assessment of possible lameness. As is already widely the case in other animal species, the use of gait analysis is a valuable aid in the objectification of lameness and the detection of subtle forms.

The article is well-organized and clear.

Nonetheless, there are some observations:

at line 115 is not clear if also the knee affected by the surgery was mono or bilateral for each rabbit;

at line 133 and 141, the authors should indicate the level of experience of veterinarians.

at line 172, the authors should specify if they had set a time limit within which the five trials could be obtained.

At line 143: table 1 orthopedic score

– grade 2 and 3 must be better defined, it is not clear what the difference between the two scores is because it is not clear what "low-grade algesia" and "middle-grade algesia" mean

-10 which means expression of pain without palpation? What parameters or clinical signs were considered?

Given the paucity of literature, the discussion fails to justify some points fully; in particular, I believe that the stimulus given to walking animals that tended not to walk may, in some way, affect their gait.

In any case, this article represents a preliminary study that can act as a forerunner to other papers.

Reviewer #2: Pressure sensing mat as an objective and sensitive tool for the evaluation of lameness in rabbits

Very interesting study on rabbit gait analysis using a pressure mat.

The syntax makes some sentences very difficult to read and the help of a professional editorial service would benefit the entire manuscript.

The caption for your figures are imbedded in the manuscript and not repeated below the figures in pages 29-34, which makes it harder to connect.

Line 26 Specify Unilateral

Line 35-37: rephrase, the sentence does not read well.

Line 38: delete "it can", conclusions following appropriate experiment and statistical analysis are definitive.

Line 62-63: Proposal: Video recording using cameras have been described and used to overcome the subjectivity and poor interobserver consistency of visual lameness scoring systems.

Line 63 delete "e.g.".

Line 64: reflective markers assigned to bone and joints: kinematics.

Line 87: Pressure mat are extensively used and their reliability is known in many species. Please cite few studies here that used this device as a gait analysis tool (e.g.: Steiner, R., Dhar, M., Stephenson, S.M., Newby, S., Bow, A., Pedersen, A. and Anderson, D.E., 2019. Biometric Data Comparison Between Lewis and Sprague Dawley Rats. Frontiers in Veterinary Science, 6, p.469.

/ Rifkin RE, Grzeskowiak RM, Mulon PY, Adair HS, Biris AS, Dhar M, Anderson DE. Use of a pressure-sensing walkway system for biometric assessment of gait characteristics in goats. PloS one. 2019 Oct 16;14(10):e0223771.)

Line 91-95: Structure this sentences in goals/aims of the study and hypothesis related to each aim.

Line 109-110: rephrase

Line 117: description the manipulation of the meniscus: luxation, tearing, folding, detachment….

Line 119-122: move this paragraph prior to the brief surgical description (line 115-118) to follow the chronological order of your experiment.

Line 129: did you measure the inter-observer agreement?

Line 131: modify "we just used in a previous study" to "as previously described [26]."

Line 133: Passive range of motion

Line 139-140: What do you mean by "not treated more than necessary"? Same comment for the inter-observer agreement?

Line 152: collocation?

Line 155: the larger areas had specified dimensions with known surface areas. Be more specific.

Lin3 196: do you mean length or duration of contact?

Line 203: Comment on the standard deviation is a result and should be presented as a result and not in the M&M section.

Line 206: it appears to have an extra parenthesis in your formula.

Line 215: the lameness scores are categorical data and cannot be presented as mean +/- SD, please explain how you handled those results.

Line 214-224: You choose to use a Wilcoxon test to compare the data of each rabbit, why did you chose to not use an ANOVA for the comparison of the data for each time points (pre-op, week 1 and week 12)?

Line 291-292: while pressure mat is certainly a more objective tool to quantify the lameness/gait of animals when compare to visual assessment, the data you present do not indicate if the leg favoring indicated by the range of both ratio pressure and ratio force was consistent for every rabbit, or if it varies per passage for each individual. If the leg favoring was consistent for one individual, why was this rabbit not excluded from the study?

One criticism that is not addressed in your discussion is the possible inconstancy for the visual lameness score evaluators.

Line 393-394: Agree, however some refinement in your article are necessary.

Table1: Why did you give a score of 10 instead of the following number (4 and 6) for the highest score? I would suggest to modify it understanding this is categorical data and not continuous data.

There is no presentation of the orthopedic score data in the result section of the manuscript. Either delete this section or include it in the result and compare with the data obtained with the pressure mat.

6. PLOS authors have the option to publish the peer review history of their article (what does this mean?). If published, this will include your full peer review and any attached files.

Reviewer #1: **Yes: **Fatone Gerardo

Reviewer #2: No

---

## [Author Response · Author response to Decision Letter 0]

7 Mar 2023

I responded to all specific reviewer and editor comments in a seperate file ("comments to reviewer"). As the editor concur with reviewers' critiques, there are no specific responses to the editor.

---

## [Decision Letter · Decision Letter 1]

26 May 2023

Pressure sensing mat as an objective and sensitive tool for the evaluation of lameness in rabbits

PONE-D-22-32303R1

Dear Dr. von der Ahe,

We’re pleased to inform you that your manuscript has been judged scientifically suitable for publication and will be formally accepted for publication once it meets all outstanding technical requirements.

Kind regards,

James H-C. Wang

Academic Editor

PLOS ONE

Additional Editor Comments (optional):

Dear Authors,

Thank you for your patience with the review process. You have adequately responded to reviewers' comments and revised the manuscript satisfactorily. The decision is to accept the manuscript on one condition that you may have an English-speaking person to help correct the English grammatical errors here and there in the manuscript.

For example, page 40, "Video recording using cameras have.." should be "Video recording using cameras has," and page 45, "physiological values were determined of each rabbit on three independent days with the sensor mat" should be changed to "Physiological values were determined for each rabbit on three independent days using the sensor mat."

Thank you,

Editor

Reviewers' comments:

Reviewer's Responses to Questions

**Comments to the Author**

1. If the authors have adequately addressed your comments raised in a previous round of review and you feel that this manuscript is now acceptable for publication, you may indicate that here to bypass the “Comments to the Author” section, enter your conflict of interest statement in the “Confidential to Editor” section, and submit your "Accept" recommendation.

Reviewer #1: All comments have been addressed

2. Is the manuscript technically sound, and do the data support the conclusions?

Reviewer #1: Yes

3. Has the statistical analysis been performed appropriately and rigorously? 

Reviewer #1: (No Response)

4. Have the authors made all data underlying the findings in their manuscript fully available?

Reviewer #1: Yes

5. Is the manuscript presented in an intelligible fashion and written in standard English?

Reviewer #1: Yes

6. Review Comments to the Author

Reviewer #1: (No Response)

7. PLOS authors have the option to publish the peer review history of their article (what does this mean?). If published, this will include your full peer review and any attached files.

Reviewer #1: **Yes: **Gerardo FATONE

---

## [Editor Report · Acceptance letter]

1 Jun 2023

PONE-D-22-32303R1 

Pressure sensing mat as an objective and sensitive tool for the evaluation of lameness in rabbits 

Dear Dr. von der Ahe:

I'm pleased to inform you that your manuscript has been deemed suitable for publication in PLOS ONE. Congratulations! Your manuscript is now with our production department. 

Kind regards, 

on behalf of

Dr. James H-C. Wang 

Academic Editor

PLOS ONE